# When the Sun Goes Down: Repairing Photometric Losses for All-Day Depth Estimation

**Madhu Vankadari**[*]
University of Oxford

**Stuart Golodetz**
University of Oxford

**Sourav Garg**
QUT, Australia

**Sangyun Shin**
University of Oxford

**Andrew Markham**
University of Oxford

**Niki Trigoni**
University of Oxford

**Abstract:** Self-supervised deep learning methods for joint depth and ego-motion estimation can yield accurate trajectories without needing ground-truth training data. However, as they typically use photometric losses, their performance can degrade significantly when the assumptions these losses make (e.g. temporal illumination consistency, a static scene, and the absence of noise and occlusions) are violated. This limits their use for e.g. nighttime sequences, which tend to contain many point light sources (including on dynamic objects) and low signal-to-noise ratio (SNR) in darker image regions. In this paper, we show how to use a combination of three techniques to allow the existing photometric losses to work for both day and nighttime images. First, we introduce a per-pixel neural intensity transformation to compensate for the light changes that occur between successive frames. Second, we predict a per-pixel residual flow map that we use to correct the reprojection correspondences induced by the estimated ego-motion and depth from the networks. And third, we denoise the training images to improve the robustness and accuracy of our approach. These changes allow us to train a single model for both day and nighttime images without needing separate encoders or extra feature networks like existing methods. We perform extensive experiments and ablation studies on the challenging Oxford RobotCar dataset to demonstrate the efficacy of our approach for both day and nighttime sequences.

## 1 Introduction

An ability to capture 3D scene structure is crucial for many applications, including autonomous driving [1], robotic manipulation [2], and augmented reality [3]. Many methods use LiDAR or fixed-baseline stereo to acquire the depth needed to reconstruct a scene, but researchers have also long been interested in estimating depth from monocular images, driven by the ubiquity, low cost, low power consumption and ease of deployment of monocular cameras. By contrast, LiDAR can be power-hungry, and stereo rigs must be calibrated and time-synchronised to achieve good performance.

Multi-view monocular depth estimation approaches have long used variable-baseline stereo over multiple images to recover depth [4, 5]. Meanwhile, progress in deep learning has opened up the additional possibility of estimating depth from a single monocular image. Deep learning methods for depth estimation can be broadly divided into two types, namely supervised methods [6, 7], and self/unsupervised methods [8, 9, 10]. Typically, supervised approaches have achieved very good results for the dataset(s) on which they are trained, but their need for ground-truth information during training has often hindered their deployment in new domains.

By contrast, self/unsupervised methods have typically adopted the use of a geometry-based loss function, inspired by the strong physical principles of traditional methods [11, 12]. This loss function is commonly referred to as the photometric or appearance loss, and is based on the assumptions that (i) the scene is static (i.e. contains no moving objects), (ii) the illumination in the scene is diffusive

---

[*]Corresponding author: `madhu.vankadari@cs.ox.ac.uk`

6th Conference on Robot Learning (CoRL 2022), Auckland, New Zealand.

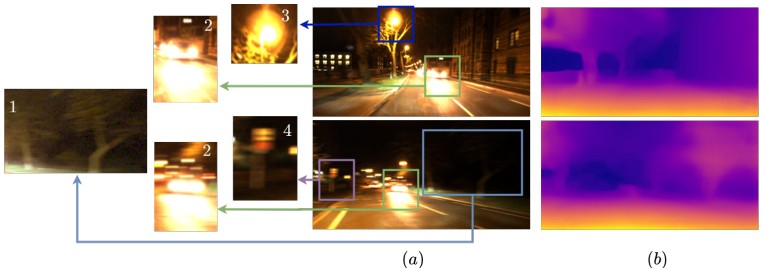

(a)                                                                              (b)

Figure 1: (a) The challenges posed by nighttime images: (1) low visibility and noise (patch enhanced for better readability); (2) moving light sources with saturating image regions; (3) point light sources; (4) extreme motion blur. (b) Despite these adverse conditions, which violate the assumptions made by the photometric loss, our method can successfully estimate accurate depth maps.

(i.e. there are no specular reflections) and temporally consistent (i.e. the pixels to which any scene point projects in any two consecutive frames have the same intensity), and (iii) the images are free of noise and occlusions [11, 12, 13, 14]. In practice, many of these assumptions are at least partly false, which can lead to errors in the estimated depth: scenes are quite likely to contain dynamic objects (e.g. cars, cyclists and pedestrians, in an outdoor driving scenario), surface materials are rarely fully diffusive, and occlusions are common. During the day, it is somewhat reasonable to assume that the illumination is moderately temporally consistent for image sequences captured outdoors, as the sun is by far the dominant light source in that case, and the light it casts changes only slowly over time; however, at night, the numerous point light sources that are typically turned on after dark (e.g. car headlights, lamp posts, etc.) can cause the illumination to change drastically from one frame to the next. At night, also, the motion blur associated with the movement of dynamic objects in the scene (including the ego-vehicle) becomes worse, owing to the longer exposure times typically used when capturing nighttime images [15, 16], and the signal-to-noise ratio of the (darker) images becomes much lower than it would be during daytime. Such issues, as illustrated in Figure 1, inhibit the straightforward use of deep networks based on photometric loss for nighttime sequences.

In this paper, we address this problem by directly targeting violations of the temporal illumination consistency, static scene and noise-free assumptions on which the photometric loss relies. As shown by our day and night results in Table 1, these three together account for much of the discrepancy in performance between daytime and nighttime. A lack of temporal illumination consistency caused by point light sources in the scene can cause pixels to be incorrectly matched between consecutive frames. To rectify this, we propose a novel per-pixel neural intensity transformation that learns to compensate for these light sources (see §3.2). Whilst conceptually straightforward, this approach is surprisingly effective, as our results in §4 demonstrate. Interestingly, they also show that it is able to operate well over wide (motion parallax) baselines, allowing us to leverage the better depth estimation performance that wider baselines offer. To correct for dynamic objects in the scene, as well as motion blur, we predict a per-pixel residual flow map (see §3.3) that we use to correct the reprojection correspondences induced by the estimated ego-motion and depth from the networks. This improves depth estimation performance at any time of day (see §4), but has additional theoretical benefits for nighttime sequences because of the greater motion blur from which they typically suffer. Lastly, we robustify our approach against noise by incorporating Neighbour2Neighbour [17], a state-of-the-art denoising module, in our photometric loss formulation (see §3.4).

## 2 Related Work

Estimating depth from images has a long history in computer vision. Several methods use either stereo images [18, 19, 20], or two or more images taken from different viewing angles [21, 22, 23]. We try to solve this problem using a single monocular image, without any constraints on the scene of interest. Various methods have addressed this problem using supervised learning [6, 7, 24, 25, 26]. However, it is infeasible to have ground-truth depth maps for training on every scene, which limits the application of these methods and helps motivate unsupervised solutions to this problem.

**Unsupervised Methods:** Garg et al. [8] proposed a geometry-based loss function to train a network in a completely unsupervised fashion using a pair of stereo images. Monodepth [27] improved this by using differentiable image warping [28] and structural similarity-based [29] image comparison loss. SfMLearner [30] used only monocular images to jointly learn depth and ego-motion. It was further improved by combining stereo and monocular losses [31, 32]. GeoNet [33] and EPC [34] learnt per-pixel optical flow maps along with depth and ego-motion to mitigate the effect of moving objects. Some GAN-based methods also exist [10, 35, 36]. Monodepth2 [37] extended Monodepth to the temporal domain, and proposed architectural changes and robust loss functions. More broadly, recent years have also seen a wide range of other advances in depth estimation, e.g. changes to the network architecture [38, 39, 40, 41], changes to the training strategy [42], the addition of extra loss functions [43], and better handling of dynamic objects [44].

**Illumination-Invariant Feature Learning**: Illumination inconsistency is a broader problem, e.g. in visual relocalisation [45] and 3D reconstruction [46]. In [47], a canonical reference image is estimated using image translation and used in a downstream pose estimation task. In [48], a sparse set of keypoints with illumination-invariant descriptors are learned from images captured at different times of day or in different weather conditions. In [49], learnt illumination-invariant features are used to compute an error metric for optimising the camera pose. Recently, many other methods [50, 51, 52, 53, 54] have targeted sparse keypoint-based relocalisation using learnt illumination-invariant feature descriptors. By contrast, in this work, we aim to solve the per-pixel depth estimation problem. Dense feature learning is also used for daytime depth estimation by [43].

**Nighttime Methods:** All the methods above are trained using photometric loss as the main supervision signal, and with an assumption of temporal illumination consistency, which is not valid at night. A few methods, such as DeFeat-Net [55], ADFA [56] and Nighttime stereo [14], have explored how to estimate depth from nighttime RGB images. DeFeat-Net [55] learns $n$-dimensional deep feature representations (assumed to be illumination-invariant) using a pixel-wise contrastive loss. The feature maps are simultaneously used along with the images for photometric loss calculation during training. ADFA [36] mimics a daytime depth estimation model by learning a new encoder that can generate 'day-like' features from nighttime images using a domain adaptation approach. Instead of feature translation as in [56], the authors in [14] propose a joint network for image translation and stereo image-based depth estimation. Recently, photometric loss has again been used in RNW [57] with an image enhancement module and a GAN-based depth regulariser. Liu et al. [58] divided the day and nighttime images into view-invariant and variant feature maps using separate encoders, and used the view-invariant information for depth estimation. All these methods either need two separate encoders for day and nighttime images [56, 58, 57], or need to learn an illumination-invariant feature space [55]. By contrast, our method learns to estimate depth in a completely self-supervised fashion, without needing stereo images, ground-truth depth or any additional feature learning.

## 3 Method

### 3.1 Baseline Method

Existing photometric loss methods typically use two networks, a depth network (or *DepthNet*) and a motion network (or *MotionNet*). The *DepthNet* takes an individual colour image as input, and is used to predict a depth image $D_t$ for each colour image $I_t$ in the input sequence. The *MotionNet* takes a consecutive pair of images $I_t$ and $I_{t+1}$ as input, and is used to output the ego-motion $T_{t,t+1}$ of the camera between them. The estimated depth and ego-motion can be used to reproject a pixel $\mathbf{u} = [u, v]^\top$ in frame $I_t$ into $I_{t+1}$ via $\dot{V}_t(\mathbf{u}) = K T_{t,t+1} D_t(\mathbf{u}) K^{-1} \dot{\mathbf{u}}$, in which $\dot{\mathbf{u}}$ is the homogeneous form of $\mathbf{u}$, $K \in \mathbb{R}^{3 \times 3}$ encodes the camera intrinsics, and $\dot{V}_t(\mathbf{u}) \in \mathbb{R}^3$ is the homogeneous form of $V_t(\mathbf{u}) \in \mathbb{R}^2$, a 2D point in the image plane of $I_{t+1}$ (which may or may not lie within the bounds of the actual image). This can be used to reconstruct an image $I'_t$ by sampling from $I_{t+1}$ around the reprojected points, using bilinear interpolation [28] to achieve a smoother result. Formally,

$$I'_t(\mathbf{u}) = \begin{cases} \text{interpolate}(I_{t+1}, V_t(\mathbf{u})) & \text{if } \mathbf{u} \in M_t \\ \mathbf{0} & \text{otherwise,} \end{cases} \qquad (1)$$

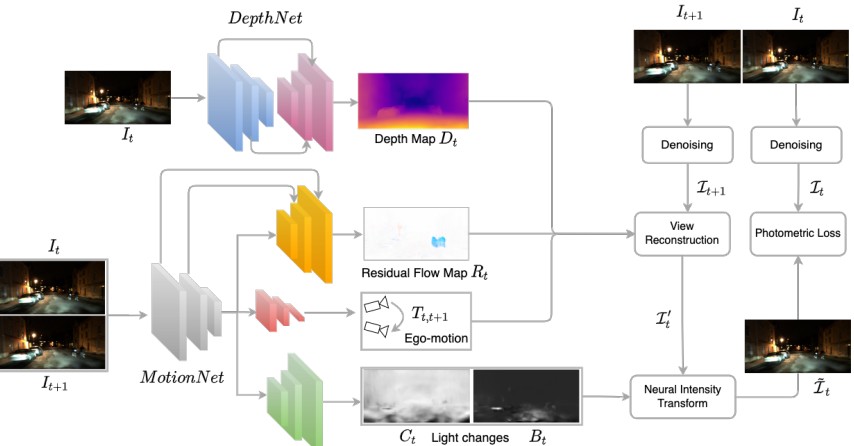

Figure 2: The architecture of our proposed method (see §3 for details).

in which $M_t = \{\mathbf{u} : \rho(V_t(\mathbf{u})) \in \Omega(I_{t+1})\}$ is the set of pixels whose reprojections into $I_{t+1}$, when rounded to the nearest pixel using $\rho$, falls within the image domain $\Omega(I_{t+1})$. The reconstructed image $I'_t$ can then be compared to the original image $I_t$ to calculate the loss values needed for training. The loss we target, namely *photometric loss*, has been used by many recent deep learning-based depth estimation techniques such as, [8, 31, 30, 36]. It is normally calculated as a convex combination of pixel-wise difference and single-scale structural dissimilarity (SSIM) [29], via

$$L_p^{(t)} = \frac{1}{|M_t|} \sum_{\mathbf{u} \in M_t} \left( \alpha \frac{1 - SSIM(I_t(\mathbf{u}), I'_t(\mathbf{u}))}{2} + (1 - \alpha) |I_t(\mathbf{u}) - I'_t(\mathbf{u})| \right). \tag{2}$$

Most existing unsupervised methods (e.g. [30, 31, 37, 38]) use this as the backbone of their formulation. To ensure a fair comparison with current nighttime state-of-the-art methods [55, 58, 57], we base our modifications in this paper on Monodepth2 [37], a commonly used baseline.

### 3.2 Lighting Change Compensation

The numerous point light sources that are typically turned on after dark (e.g. car headlights, lamp posts, etc.) can cause the illumination of a scene to change significantly from frame $I_t$ to frame $I_{t+1}$. Moreover, when a light source moves with the camera (e.g. car headlights), this can lead to large holes in the estimated depth in front of the ego-vehicle [55, 58]. To compensate for the illumination changes, we estimate a per-pixel transformation that, when applied to $I_{t+1}$, can mitigate the changes in lighting that have occurred since $I_t$. We draw some inspiration from [59, 60, 61], which use a single whole-image transformation based on two scalar values to compensate for the difference in exposure time between a pair of images, as this creates roughly uniform intensity changes over the entire image. However, in our case, the intensity changes are far from uniform over the image, owing to both the motions of the ego-vehicle and other objects in the scene, and the distances between the ego-vehicle and static point light sources. For this reason, we propose a per-pixel formulation here.

Our approach starts by passing the features produced by the last convolutional layer of the *Motion-Net* through a *lighting change decoder*[2] to estimate two per-pixel change images, $C_t$ and $B_t$ (see Figure 2). These (respectively) aim to capture the per-pixel changes in contrast (scale) and brightness (shift) that have occurred between the two input frames. As shown in Figure 4, the brightness image $B_t$ broadly captures the extra light added to the image by e.g. vehicle headlights, and the contrast image $C_t$ broadly captures the changes in ambient light due to the motion of the ego-vehicle towards or away from point light sources such as street lamps. We use these images to transform the reconstructed image $I'_t$ via $\tilde{I}_t = C_t \odot I'_t + B_t$, in which $\odot$ denotes the Hadamard product.

---

[2]Similar to the *DepthNet* decoder, but without skip connections. See supplementary material for details.

### 3.3 Motion Compensation

The standard photometric loss makes use of correspondences $\{\mathbf{u} \leftrightarrow V_t(\mathbf{u}) : \mathbf{u} \in M_t\}$ between consecutive frames that have been established via reprojection, based on the estimated ego-motion and depth. Assuming that (i) the ego-motion and depth have been estimated well, (ii) the scene is static, and (iii) there is minimal motion blur, these correspondences will broadly match those that would have been established had we used the ground-truth optic flow $\Phi_t(\cdot)$ from frame $t$ to frame $t+1$. However, if objects move with respect to the background scene, or anything visible in the image moves with respect to the ego-camera (which can cause motion blur), then these correspondences may be incorrect. To correct them, we predict a residual flow map $R_t$, such that for each pixel $\mathbf{u}$, $R_t(\mathbf{u}) \in \mathbb{R}^2$ is an estimate of $(\mathbf{u} + \Phi_t(\mathbf{u})) - V_t(\mathbf{u})$. We can then add $R_t(\mathbf{u})$ to $V_t(\mathbf{u})$ to obtain a potentially more accurate correspondence for use in reconstructing $I'_t$ via Equation 1.

Some methods [34, 33, 44] already exist that predict residual flow for daytime images by using a separate encoder-decoder network or computationally intensive image warping-based bilinear interpolation for supervision. By contrast, we estimate residual flow using an efficient sparsity-based formulation. This involves introducing a *residual flow decoder* that takes the features of the final convolutional layer of the *MotionNet* as input and the features of previous layers in the *MotionNet* via skip connections, and outputs residual flow maps $\{R_{t,s} : s \in \{0, 1, 2, 3\}\}$ at four different scales (each $R_{t,s}$ has a width and height that is $1/2^s$ that of $I_t$, and $R_{t,0} \equiv R_t$).

There is no direct supervision available to learn the residual flow maps. For this reason, we choose instead to encourage sparsity in the residual flow estimates, so that the estimated depth and ego-motion can explain the majority of the scene, and the left-over can be explained by the residual flow maps. To achieve this, we adopt the sparsity loss from [44], i.e.

$$L_r^{(t)} = \sum_{s=0}^{3} \langle |R_{t,s}| \rangle / 2^s \sum_{\mathbf{u} \in \Omega(I_{t,s})} \sqrt{1 + |R_{t,s}(\mathbf{u})| / \langle |R_{t,s}| \rangle}, \tag{3}$$

in which $I_{t,s}$ is a downsampled version of $I_t$ at scale $s$, and $\langle |R_{t,s}| \rangle$ is the spatial average of the absolute residual flow map $|R_{t,s}|$. By contrast with [44], here we introduce a normalising factor of $1/2^s$ at each scale, since the original loss was for scene flow, where the flow magnitude is independent of the resolution of the flow maps, which is not the case for the 2D residual flow we consider.

### 3.4 Image Denoising

Image noise is yet another key factor that affects the performance of the photometric loss, especially in darker regions of the image that typically have a low SNR. Handling this noise is crucial, as photometric loss is the only training signal. To denoise the images, we used Neighbour2Neighbour [17], a state-of-the-art unsupervised model trained on ImageNet with zero-mean Gaussian noise. The standard deviation was varied from 5 to 50 during training. We denoise the images before feeding them to the network. This can either be done at both training and test times, or solely at training time (for calculating the loss). We chose the latter, as the former has two potential disadvantages: (i) it can significantly add to the computational burden at runtime, slowing down the depth estimation, and (ii) any errors in the denoising process can lead to downstream errors in the depth maps, even though the depth estimation model itself might have been trained well. By contrast, restricting denoising to training time has the advantage of allowing us to make the depth and motion networks robust to noise by training them on the original noisy images, but supervising with the denoised images.

### 3.5 Full Pipeline

We can now formulate our full pipeline, which takes two consecutive images $I_t$ and $I_{t+1}$, as

$$
\begin{aligned}
&D_t = \mathcal{D}(I_t), \ f_n = \mathcal{ME}_{1:n}([I_t, I_{t+1}]), \ T_{t,t+1} = \mathcal{MD}(f_N) \\
&R_t = \mathcal{RFD}(\{f_n : 1 \le n \le N\}), \ (C_t, B_t) = \mathcal{LCD}(f_N) \\
&\mathcal{I}_t = \mathcal{DN}(I_t), \ \mathcal{I}_{t+1} = \mathcal{DN}(I_{t+1}), \ \mathcal{I}'_t = \texttt{reconstruct}(\mathcal{I}_{t+1}, V_t + R_t), \ \tilde{\mathcal{I}}_t = C_t \odot \mathcal{I}'_t + B_t \\
&L_p^{(t)} = \frac{1}{|M_t|} \sum_{\mathbf{u} \in M_t} \left( \alpha \frac{1 - SSIM(\mathcal{I}_t(\mathbf{u}), \tilde{\mathcal{I}}_t(\mathbf{u}))}{2} + (1 - \alpha) \left| \mathcal{I}_t(\mathbf{u}) - \tilde{\mathcal{I}}_t(\mathbf{u}) \right| \right),
\end{aligned}
\tag{4}
$$

where $\mathcal{D}$ is the *DepthNet*, $\mathcal{ME}_{1:n}$ denotes the first $n$ layers of the $N$-layer *MotionNet* encoder, $\mathcal{MD}$ is the *MotionNet* decoder, $\mathcal{RFD}$ is the residual flow decoder, $\mathcal{LCD}$ is the lighting change decoder, and $\mathcal{DN}$ is the denoiser [17]. The `reconstruct` function reconstructs $\mathcal{I}'_t$ as per Equation 1, and $V_t$ denotes the reprojected pixels from $I_t$ to $I_{t+1}$. At frame $t$, the *DepthNet* is applied to $I_t$ to estimate a per-pixel depth map $D_t$, and the *MotionNet* is applied to both $I_t$ and $I_{t+1}$ to estimate the ego-motion $T_{t,t+1}$ from frame $t$ to frame $t+1$. During training, the features from the last convolutional layer of the *MotionNet* are passed through the *lighting change decoder* to estimate the per-pixel lighting changes that have occurred between the frames (see §3.2). A separate *residual flow decoder* takes both these last layer features and the features of earlier layers in the *MotionNet* (via skip connections) to estimate a per-pixel residual flow map $R_t$ (see §3.3) that we use to correct the reprojection correspondences induced by the estimated ego-motion and depth from the networks. To calculate the photometric loss, we (i) denoise $I_t$ and $I_{t+1}$ with a state-of-the-art denoising module [17] to produce $\mathcal{I}_t$ and $\mathcal{I}_{t+1}$; (ii) reconstruct $\mathcal{I}'_t$ from $\mathcal{I}_{t+1}$ using reprojection correspondences that have been corrected using the residual flow; (iii) correct $\mathcal{I}'_t$ for the estimated lighting changes between the frames to produce $\tilde{\mathcal{I}}_t$; and then (iv) calculate the usual photometric loss between $\tilde{\mathcal{I}}_t$ and $\mathcal{I}_t$.

**Making the Pipeline Bidirectional.** Monodepth2 [37] calculates its photometric loss not only forwards, from $I_t$ to $I_{t+1}$, but also backwards, from $I_t$ to $I_{t-1}$, before combining the losses. This allows us to use the idea of minimum reprojection error to account for occluded pixels, and so we do the same. We also adopt the auto-masking losses from Monodepth2 [37], which we call $L_a^{(t)}$, as even though our method can cope with moving objects, it is very difficult to use parallax to disentangle the motion of objects that are moving in the same direction and at the same speed as the ego-vehicle. We further include the commonly used edge-aware gradient smoothing loss $L_g^{(t)}$ [27] to maintain spatial smoothness over the estimated depth maps. Our final loss $L^{(t)}$ then becomes

$$L^{(t)} = \min\left(L_{p-}^{(t)}, L_{p+}^{(t)}, L_{a-}^{(t)}, L_{a+}^{(t)}\right) + \lambda_r\left(L_{r-}^{(t)} + L_{r+}^{(t)}\right) + \lambda_g L_g^{(t)}, \tag{5}$$

in which $+/-$ denote the forward/backward versions of the losses, and $\lambda_r, \lambda_g \in \mathbb{R}$ are the weights.

## 4 Experiments

In §4.1, we compare our depth estimation performance to a number of state-of-the-art approaches in a variety of different daytime and/or nighttime contexts. In §4.2, we present a study on the effect of parallax to help explain the importance of our neural intensity transformation module. Finally, in §4.3, we perform an ablation study to analyse the contributions made by the three individual components of our approach. Further experiments can be found in the supplementary material.

### 4.1 Depth Evaluation

We compare with 4 state-of-the-art unsupervised monocular methods: Monodepth2 [37], DeFeat-Net [55], ADDS-Depth-Night [58] and RNW [57] (see Figure 3 and Table 1). The quantitative evaluation uses the error and accuracy metrics from [6], as detailed in the supplementary material. We tested our model with 3 different data variations: day only ($d$), night only ($n$), and a mix of day and night ($d\&n$). Monodepth2 [37] can be trained with all 3 configurations, but has already been outperformed by DeFeat-Net [55] in the $d\&n$ setting. For the $d$ and $n$ settings, we outperform it by a significant margin in both error and accuracy (see Table 1). DeFeat-Net [55] and ADDS-Depth-Night [58] were originally trained with a $d\&n$ configuration. We evaluated the pre-trained models they released on our test split. Our method outperforms both methods by a significant margin on the nighttime sequences (see Table 1). Please note that we do not use any additional feature representation-based losses as used in DeFeat-Net [55], or paired day and night images as used in ADDS-Depth-Net [58]. RNW [57] is also built on Monodepth2, but targets nighttime data only. As per Figure 3, our depth estimation results are sharper and better able to preserve edges than the competing methods. We also found that using a longer baseline improves depth estimation performance. However, naïvely using a wider baseline without also using our neural intensity transform can lead to a severe decrease in accuracy, particularly for nighttime images.

| Test | Method | Train | Abs. Rel. | Sq. Rel. | RMSE | Log RMSE | $\delta < 1.25$ | $\delta < 1.25^2$ | $\delta < 1.25^3$ |
|---|---|---|---|---|---|---|---|---|---|
| *Day* | Monodepth2 [37] | d | 0.219 | 4.525 | 7.641 | 0.285 | 0.679 | 0.862 | 0.930 |
| | Ours | d | **0.191** | **1.710** | **6.158** | **0.253** | **0.713** | **0.904** | **0.962** |
| | DeFeat-Net [55] | d & n | 0.247 | 2.980 | 7.884 | 0.305 | 0.650 | 0.866 | 0.943 |
| | RNW [57] | d & n | 0.297 | 2.608 | 7.996 | 0.359 | 0.431 | 0.773 | 0.930 |
| | ADDS-Depth-Night [58] | d & n | 0.239 | 2.089 | 6.743 | 0.295 | 0.614 | 0.870 | 0.950 |
| | Ours | d & n | **0.176** | **1.603** | **6.036** | **0.245** | **0.750** | **0.912** | **0.963** |
| *Night* | Monodepth2 [37] | n | 0.453 | 21.310 | 11.420 | 0.444 | 0.700 | 0.873 | 0.930 |
| | RNW MCIE + SBM [57] | n | 0.350 | 7.934 | 8.994 | 0.407 | 0.674 | 0.861 | 0.922 |
| | Ours | n | **0.186** | **1.656** | **6.288** | **0.248** | **0.728** | **0.919** | **0.969** |
| | DeFeat-Net [55] | d & n | 0.334 | 4.589 | 8.606 | 0.358 | 0.586 | 0.827 | 0.911 |
| | ADDS-Depth-Night [58] | d & n | 0.287 | 2.569 | 7.985 | 0.339 | 0.490 | 0.816 | 0.946 |
| | RNW [57] | d & n | 0.185 | 1.710 | 6.549 | 0.262 | 0.733 | 0.910 | 0.960 |
| | Ours | d & n | **0.174** | **1.637** | **6.302** | **0.245** | **0.754** | **0.915** | **0.964** |

Table 1: A quantitative comparison of our method. The results of Monodepth2 [37] are reported after retraining it. Those of DeFeat-Net [55] and ADDS-Depth-Night [58] are reported using the checkpoints from their public repositories. The evaluation uses a maximum depth of 50m. Underlined methods use daytime images as main supervision or for regularisation losses.

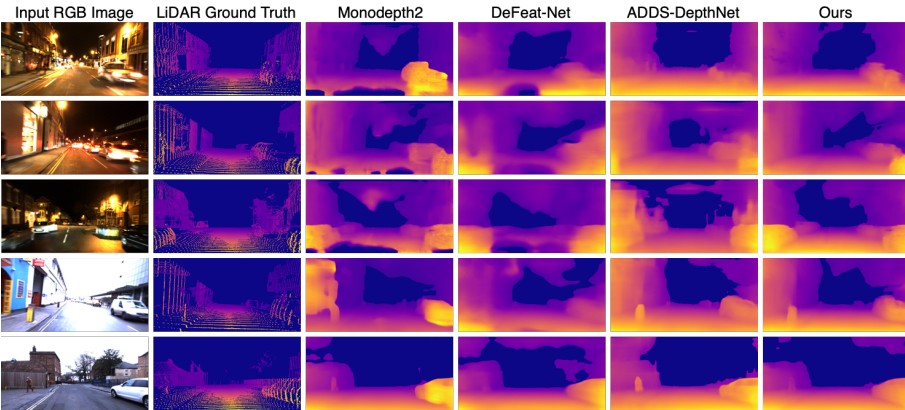

Figure 3: A qualitative comparison of our proposed method with the state of the art.

## 4.2 Effect of Parallax

To better understand how depth estimation performance is affected by increasing the average parallax (metric separation) between the images we use to calculate the photometric loss, we constructed a new nighttime training split by increasing the intra-triplet stride (see supplementary material) to 2, which increased the average parallax between the images from 0.353m to 0.706m. Without our neural intensity transformation, the depth estimation performance significantly decreased compared to the original nighttime training split (see the difference between the RMSEs of the baseline in the top and bottom parts of Table 2). A key cause of this in night images is likely the headlights of the ego-vehicle, which can cause the pixel intensities to change drastically between frames. However, with our neural intensity transformation, the depth estimation performance was found to instead increase, which we hypothesise to be because by compensating for the lighting changes, we make it possible to exploit the stronger supervision that can be offered by a wider baseline.

## 4.3 Ablation Study

**Lighting Change Compensation.** In Figure 4(a), we show several reference images and their lighting change maps. The intensity changes are non-uniform, so we cannot use the existing correction approaches from [59, 60, 61]. We also observe that our method is able to clearly disentangle both the changes in ambient light resulting from movement towards/away from point light sources (captured by $C_t$) and the additional light added to the road pixels in the images by the ego-vehicle headlights (captured by $B_t$). Our neural intensity transform significantly reduces the RMSE compared to the baseline (see Table 2), and is also able to fill in holes in front of the ego-vehicle (see Figure 3).

| Stride | Method | Abs. Rel. | Sq. Rel. | RMSE | Log RMSE | $\delta < 1.25$ | $\delta < 1.25^2$ | $\delta < 1.25^3$ |
|---|---|---|---|---|---|---|---|---|
| 1 | Baseline | 0.266 | 5.647 | 6.305 | 0.331 | 0.759 | 0.9013 | 0.947 |
| | w/ NIT | 0.190 | 1.824 | 4.848 | 0.257 | 0.763 | 0.919 | 0.965 |
| | w/ Denoising | 0.163 | 1.256 | 4.193 | 0.224 | 0.801 | 0.935 | 0.973 |
| | Full Model | 0.154 | 1.174 | 4.120 | 0.216 | 0.811 | 0.939 | 0.976 |
| 2 | Baseline | 0.602 | 63.914 | 14.726 | 0.467 | 0.785 | 0.902 | 0.939 |
| | w/ NIT | 0.169 | 1.727 | 4.693 | 0.236 | 0.812 | 0.929 | 0.967 |
| | Full Model | 0.131 | 0.926 | 3.731 | 0.188 | 0.852 | 0.949 | 0.980 |

Table 2: Ablation study showing the importance of different modules in our system where NIT stands for neural intensity transformation. The maximum evaluation depth was set to 30m for this study. 'Stride' denotes intra-triplet stride (see supplementary material).

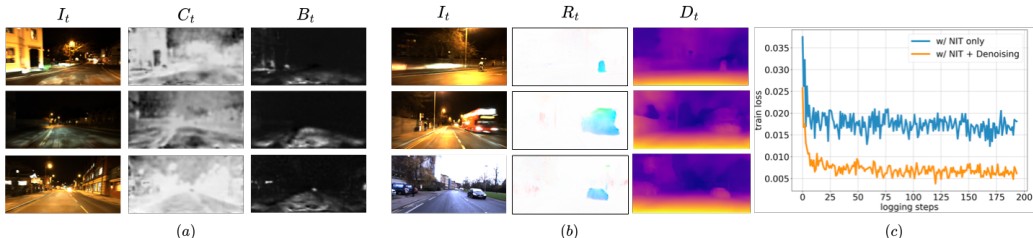

Figure 4: Visualisations of (a) estimated light changes; (b) residual flow and estimated depth; (c) the effects of denoising on the training loss over time. Please refer to §4.3 for more details.

**Motion Compensation.** In Figure 4(b), we show several reference images and their residual flow and depth maps. In the second column, one can clearly see that our method is able to distinguish pixels on moving objects such as cars and pedestrians from static pixels. This effect can be observed for both daytime and night-time images, showcasing the generality of our approach through a single unified training pipeline. In Table 2, it can be seen that correcting the reprojection correspondences using the residual flow map we predict leads to a significant improvement in accuracy.

**Image Denoising.** Denoising the images while calculating the training loss should ideally reduce the ambiguity in establishing pixel correspondences between the images, giving a robust supervision signal for training our system and thereby achieving lower errors and higher accuracy. This effect can be clearly seen in the training error plot shown in Figure 4(c), where we compare our baseline+NIT model with and without denoising. The denoising results in much more accurate depth maps, improving both the RMSE and accuracy metrics as shown in Table 2.

## 5   Limitations

Like most stereo approaches (variable-baseline like ours, or fixed-baseline with a rigid stereo rig), our method struggles to preserve the detail of distant parts of the scene because of limited parallax. It also struggles to recover structural detail from very dark image regions (e.g. see Figure 1(b)). Furthermore, we currently approximate the relationship between pixel intensities and light intensity as linear for simplicity. This can be better approximated by using the inverse camera-response function. Finally, whilst our method copes with most motion patterns, it can struggle to estimate the motion of objects moving with the same speed as the ego-vehicle and in the same direction.

## 6   Conclusions

In this paper, we propose a self-supervised method to learn a single model to estimate depth maps from monocular day and nighttime RGB images. By compensating for the illumination changes that can occur from one frame to the next, we enable accurate nighttime depth estimation in non-uniform lighting conditions. Moreover, by predicting per-pixel residual flow and using it to correct the reprojection correspondences induced by the estimated ego-motion and depth, we improve our method's ability to cope with both moving objects in the scene and motion blur. Finally, by denoising the input images prior to calculating the photometric loss, we improve the loss's ability to provide a strong supervision signal, making the entire system more robust and accurate.

**Acknowledgments**

This work was supported by AWS via the Oxford-Singapore Human-Machine Collaboration Programme, and EPSRC via ACE-OPS (EP/S030832/1). The authors would also like to thank the anonymous reviewers for their helpful comments.

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
