# OpenReview forum: "When the Sun Goes Down: Repairing Photometric Losses for All-Day Depth Estimation"
_robot-learning.org/CoRL/2022/Conference — CoRL 2022 Poster_

### Official Review · Reviewer_iBSx · 2022-07-22

**Originality:** Fair
**Technical Quality:** Very Good
**Clarity Of Presentation:** Very Good
**Impact:** 3

**Recommendation:**

Weak Reject: I recommend rejecting the paper, but will not argue for my recommendation if the majority of other reviewers have a different opinion.

**Summary:**

The paper introduced an improved network for depth estimation, where the photometric loss is intensified for varying illumination conditions. Specifically, the network is based on an encoder-decoder DepthNet for depth prediction and a MotionNet for pose estimation. The authors proposed three modules to improve the photometric loss for robustness against illumination change (e.g., day to night) and moving objects. They augment the network to predict the contrast and brightness parameters for each pixel to mitigate the lighting change, and predict residual flows to detect moving objects; additionally, they use an image denoising network to remove noise from the images. The evaluation shows the improvements on depth prediction.

**Issues:**

- Section 3.5 is not very intuitive. It might be easy to follow if the equations are replaced with descriptions.
- The metrics used in the tables need explanation and unit.

**Quality Of The Limitations Section:**

Limitations are addressed clearly

**Reviewer Expertise:**

2: The reviewer is willing to defend the evaluation, but it is quite likely that the reviewer did not understand central parts of the paper

**Robotics Focus:**

Highly relevant to robotics but no hardware experiments

**Strengths And Weaknesses:**

Strength
- The paper is well-written, and the method is clearly stated and technically sound. The comparison against existing work and the ablation study clearly shows the improvements with the intensified photometric loss.

Weakness
- For novelty, it is an incremental work modified from existing methods. Nevertheless, the performance is good.
- The reviewer thought the proposed method is also learning-based visual odometry instead of just predicting depth since the ego-motion is also estimated. However, the evaluation is purely based on depth. The reviewer is also interested to see the evaluation of ego-motion.


**Summary Of Recommendation:**

It is a good system work for depth estimation in challenging conditions, the evaluation clearly shows the improvement. But the reviewer is not sure about the novelty of the method.

---

> ### Author Response · Authors · 2022-08-26
> **Response to Reviewer iBSx**
>
> Q1: Novelty
>
> A1: Our method is the first unsupervised depth estimation system that can be trained purely using any combination of night-time/day-time/both RGB images without needing additional losses/data. Instead of relying on or reinforcing the existing photometric losses with GAN-based or additional feature-based losses, here, we aim to address the fundamental problems associated with the dynamic night-time image data such as illumination inconsistency, blurry moving objects, and low SNR. With simple yet powerful modifications to a relatively simpler day-time baseline architecture with ResNet-18 network, monodepth2, we are able to outperform methods such as RNW and DeFeatNet which use ResNet-50, demonstrating the effectiveness of the contributions made in this paper. We believe that this paper is the first step towards addressing the problem of night-time depth estimation using an approach grounded in physics rather than using computationally heavy and brittle architectures like GANs. In addition to the methodological contributions, we provide a standard benchmark split for the Oxford RobotCar dataset.
>
> Q2: Section 3.5 is not very intuitive. It might be easy to follow if the equations are replaced with descriptions.
>
> A2: We will add a detailed description of the entire pipeline to the revised version of the paper.
>
> Q3: The metrics used in the tables need explanation and unit.
>
> A3: We will add this to the revised version of the paper. These metrics are taken from Eigen et al., NeurIPS,2014.
>
>
> Q4: The reviewer thought the proposed method is also learning-based visual odometry instead of just predicting depth since the ego-motion is also estimated. However, the evaluation is purely based on depth. The reviewer is also interested to see the evaluation of ego-motion.
>
> A4: We had evaluated and presented ego-motion results in our submitted version, please see Section 3 in the supplementary. Although ego-motion estimation is part of our depth estimation pipeline (similar to existing works), it is not the main focus of our work and is not necessarily needed during inference (thus saving computation time). Nevertheless, our method achieves competitive performance on this task compared to existing works (please see Table 3 and Figure 2 in the supplementary).
>
> References:
>
> Eigen D, Puhrsch C, Fergus R. Depth map prediction from a single image using a multi-scale deep network. Advances in neural information processing systems. 2014;27.

---

> > ### Comment · Reviewer_iBSx · 2022-08-27
> > **Thank you for your explanation and clarification.**
> >
> > Thank you for your explanation and clarification.

---

### Official Review · Reviewer_JhM1 · 2022-07-25

**Originality:** Good
**Technical Quality:** Good
**Clarity Of Presentation:** Very Good
**Impact:** 4

**Recommendation:**

Weak Accept: I recommend accepting the paper, but will not argue for my recommendation if the majority of other reviewers have a different opinion.

**Summary:**

This paper proposes three techniques to improve self-supervised monocular depth estimation for both day and nighttime images: (i) per-pixel neural intensity transformation for lighting change compensation, (ii) per-pixel residual flow map motion compensation, and (iii) image denoising for robustness improvement. Several constructive insights are introduced in the paper to address the problem of performance degradation in scenarios when the assumption of photometric losses is violated. Experimental results and ablation studies are performed on the Oxford RobotCar dataset.

**Issues:**

Authors are encouraged to address the few issues mentioned above to improve clarity and readability of this paper.

**Quality Of The Limitations Section:**

Additional details required

**Reviewer Expertise:**

4: The reviewer is confident but not absolutely certain that the evaluation is correct

**Robotics Focus:**

Relevant but unlikely to deploy to hardware in near future

**Strengths And Weaknesses:**

Improving the depth estimation accuracy of nighttime images is an important problem, and the authors provide three good ideas. This paper presents its novel contributions in a coherent and clear manner; the methodological details for addressing the temporal illumination consistency, static scene, and noise-free assumptions make sense. The experimental results seem convincing as well.

However, there are several issues and weakness in this paper, which are listed below:

    * The authors lack elaboration on some technical details. For example, "lighting change decoder" has been mentioned in Line 148 but no corresponding network structure is provided.

    * In section 3.4, the authors claimed the denoising model used solely at training time is better than used at both training time and test time due to the better robustness. However, no experimental evidence is shown in the paper to back this claim.

    * Insights for practical implementation of the proposed methods for real-time robot vision or robot learning - are missing. At-least some on-device feasibility analyses / computational reasoning should have been included.

    * This paper lacks a detailed description of the tables and figures that demonstrate the experiment results. For example, “NIT“ in Table 2 is not clearly defined, and the experimental results of Figure 4 are not well explained.

**Summary Of Recommendation:**

This paper makes some good contributions toward addressing photometric loss assumptions in low-light conditions and night-time images. Although their practical implementation for real-time robot vision are not straight-forward, but the proposed concepts can be useful to the community.

---

> ### Author Response · Authors · 2022-08-26
> **Response to Reviewer JhM1**
>
> Q1: The authors lack elaboration on some technical details. For example, "lighting change decoder" has been mentioned in Line 148 but no corresponding network structure is provided
>
> A1: Our light-change decoder is similar to the DepthNet decoder without any skip connections from the motion-net encoder. Thanks for pointing it out, we will add the architecture details to the revised supplementary material.
>
>
> Q2: In section 3.4, the authors claimed the denoising model used solely at training time is better than used at both training time and test time due to the better robustness. However, no experimental evidence is shown in the paper to back this claim.
>
> A2: This is in reference to L190-196, where our original misleading phrasing led to this confusing claim, which has been thankfully spotted by the reviewer. We refer to our proposed pipeline in Figure 2, where it can be seen that the denoising module is not part of DepthNet but is only for loss computation during training, while DepthNet is what is used during testing. Thus, an ablation study with and without the denoising module during testing is not possible.
>
> The confusion arises from our discussion (L190-196) of an alternative architecture where a denoising module could have been used before passing images to DepthNet, where it would have been a must to use it during both training and testing. Since this alternative architecture will have unnecessary computational overhead for testing and deployment (regardless of performance improvement), we did not consider it worth exploring as our proposed architecture is already leveraging denoising in an efficient way to improve performance.
> We will revise this text in the paper to clearly present our architecture and avoid this confusion.
>
>
> Q3: Insights for practical implementation of the proposed methods for real-time robot vision or robot learning - are missing. At-least some on-device feasibility analyses / computational reasoning should have been included.
>
> A3: Great suggestion! We have now included a subsection on Compute Time Analysis in the revised version, as below:
>
> Here, we discuss the test time computational complexity of our inference network (DepthNet), which is composed of a ResNet-18 encoder and a U-Net-style decoder. The total number of parameters of this model is 14.84M and computational complexity, measured in GMACs (Giga Multiply Accumulate per second), is 8.58. For a batch size of 1, the model runs at 231 fps (0.00423 seconds) on Nvidia A10 GPU and 34 fps (0.02887 seconds) on Intel(R) Xeon(R) Silver 4314 CPU @ 2.40GHz. These numbers were estimated by averaging over 300 runs. We used ptflops (https://github.com/sovrasov/flops-counter.pytorch) library to estimate GMACs. Below, we compare our model with an existing state-of-the-art method RNW (Wang K et al, CVPR, 2021), demonstrating significant computational advantage over the state-of-the-art.
>
> RNW -  33.12 GMACs with 32.84 Million parameters
>
> Ours -   8.58 GMACs with 14.84 Million parameters
>
> Q4: This paper lacks a detailed description of the tables and figures that demonstrate the experiment results. For example, “NIT“ in Table 2 is not clearly defined, and the experimental results of Figure 4 are not well explained.
>
> A4: We call the process of compensating for the light changes as Neural Intensity Transformation (NIT). Thanks for pointing it out, we will add this description to the table caption. We will update the Figure 4 caption to briefly explain the results as discussed in the corresponding Section 4.3.
>
> References:
>
> Wang, Kun, et al. "Regularizing nighttime weirdness: Efficient self-supervised monocular depth estimation in the dark." Proceedings of the IEEE/CVF International Conference on Computer Vision. 2021.

---

### Official Review · Reviewer_YCUH · 2022-07-31

**Originality:** Very Good
**Technical Quality:** Excellent
**Clarity Of Presentation:** Excellent
**Impact:** 4

**Recommendation:**

Strong Accept: I recommend accepting the paper and will argue for my recommendation even if other reviewers hold a different opinion.

**Summary:**

This paper describes an approach to the self-supervised training of deep networks for joint depth and ego-motion estimation that are capable of accurate all-day (i.e. day *and* night) prediction.  The main technical contribution is to identify three assumptions underpinning the use of the photometric loss for self-supervision that are significantly violated during nighttime operation (temporal illumination consistency, static scenes, and absence of noise and occlusions), and then show how to modify standard architectures for monocular depth and ego-motion estimation to correct for these sources of error.  The result is a single network architecture that can be trained and deployed on any combination of day and nighttime image sequences.  The experimental results provide convincing evidence that the proposed system significantly outperforms comparable baselines on both day and nighttime applications, including prior works specifically designed for nighttime use.

**Issues:**

 Given that this paper is in my opinion already quite strong, I have only a few (minor) comments:

* While it is stated in-line earlier in the paper, I think it might be good to include the definition of Vt(u) in the statement of the full pipeline in equation (4) for easy reference.

* Similarly, it would be good to explicitly state in the paper what are the performance measures reported in Tables I and II.  (RMSE and log RMSE are standard statistical measures, but “Abs. Rel.”, “Sq. Rel”, and the various delta measures are I think less so.  (Also, what is the significance of 1.25, 1.25^2, and 1.25^3 as thresholds?)

* It would also be good to mention how the weighting parameters lambda_r and lambda_g appearing in the loss in equation (5) are chosen.


**Quality Of The Limitations Section:**

Limitations are addressed clearly

**Reviewer Expertise:**

3: The reviewer is fairly confident that the evaluation is correct

**Robotics Focus:**

Sufficient demonstration on hardware

**Strengths And Weaknesses:**

In my opinion the paper’s greatest strength is that the technical approach is explicitly grounded in the physics and geometry of image formation.

For example, the paper motivates the development of the lighting change compensation module by considering the validity of the “temporal illumination consistency” assumption in daytime operation (where most of the ambient illumination comes from the sun, which is well-approximated as a static source) versus nighttime operation (where much of the ambient illumination comes from traffic lights, streetlamps, and headlights of other vehicles, which are bright point sources that move over time relative to a moving camera).  The proposed illumination compensation method is a simple yet effective linear model based upon a predicted *contrast* map (which is intended to capture *global* changes in illumination caused by motion of the vehicle towards / away from ambient sources such as streetlights) and a predicted *brightness* map (which accounts for the presence of potentially moving *point* sources, such as vehicle headlights).  Similar analyses lead to the motion correction approach (designed to account for motion blur and / or the motion of objects in the scene, such as vehicles) and the denoising approach used to pre-process the training images.

The design of the proposed approach is thus well-justified, physically grounded, and readily interpretable, all of which are highly desirable properties.

Furthermore, because the approach is based upon accounting for violations of the assumptions underpinning the photometric loss (which can occur in *day* as well as night operation, although typically to a lesser degree), the resulting corrections serve to improve the performance of the overall system in both day *and* night operation, as shown by the experiments in Section 4.  Indeed, those results show that the proposed approach provides a single architecture and training procedure that outperforms *all* tested baselines across *all* combinations of day and nighttime training and test sequences.

The paper thus combines a principled design approach with a clear and convincing experimental demonstration of the method’s advantages over prior art.

I did not find any significant weaknesses in the paper (the method of course has its limitations, but in my view these are adequately addressed by the paper itself).




**Summary Of Recommendation:**

I recommend accepting this paper on the basis that it addresses an important problem in monocular depth and ego-motion estimation (extending these methods to operate reliably at night), proposes a thoughtfully-designed and well-justified technical approach, and provides clear and convincing experimental evidence of the method’s advantages over prior work, in both day *and* nighttime operation.

---

> ### Author Response · Authors · 2022-08-26
> **Response to Reviewer YCUH**
>
> Q1: While it is stated in-line earlier in the paper, I think it might be good to include the definition of Vt(u) in the statement of the full pipeline in equation (4) for easy reference.
>
> A1: We will do this in the revised version of the paper.
>
> Q2: Similarly, it would be good to explicitly state in the paper what are the performance measures reported in Tables I and II. (RMSE and log RMSE are standard statistical measures, but “Abs. Rel.”, “Sq. Rel”, and the various delta measures are I think less so. (Also, what is the significance of $1.25$, $1.25^2$, and $1.25^3$ as thresholds?)
>
> A2: We will add these details to the revised version of the paper. All the metrics used in those tables were originally used in Eigen et al., NeurIPS, 2014; we will explicitly define all these metrics in the experimental setup.
>
> Significance of thresholds for delta measure:
>
> The ‘delta measure’ is the inlier ratio of pixels that do not over- or under-estimate depth beyond a certain margin. These margins can be any thresholds; in our case, we use three thresholds that cover an increasing relative change as $t$, $t^2$, and $t^3$. Since we use a max operation over the $\frac{D_{t} }{D^*_{t}}$ and $\frac{D^*_{t} }{D_{t}}$, $t > 1$ and in our case set to $1.25$, which accounts for both overshooting $(\frac{D_{t} }{D^*_{t}})$ and undershooting $(\frac{D^*_{t} }{D_{t}})$. As both RMSE and Sq-Rel Error metrics are affected by outliers, using the aforementioned thresholds-based delta measure provides insights into the distribution of depth errors.
>
> Q3: It would also be good to mention how the weighting parameters lambda_r and lambda_g appearing in the loss in equation (5) are chosen.
>
> A3: Thanks for pointing this out. The residual sparsity regularization weight $\lambda_r$ was chosen by ablating three different weight values: $5e-2$, $5e-3$, and $1e-3$, where $1e-3$ was found to work the best in terms of RMSE metrics as evaluated on the validation set. The gradient smoothening weight $\lambda_g$ was directly borrowed from the baseline method Monodepth2. We will add this in the revised version of the paper.
>
> References:
>
> Eigen D, Puhrsch C, Fergus R. Depth map prediction from a single image using a multi-scale deep network. Advances in neural information processing systems. 2014;27.

---

### Official Review · Reviewer_BWdU · 2022-08-01

**Originality:** Good
**Technical Quality:** Good
**Clarity Of Presentation:** Very Good
**Impact:** 3

**Recommendation:**

Weak Accept: I recommend accepting the paper, but will not argue for my recommendation if the majority of other reviewers have a different opinion.

**Summary:**

This paper presents three contributions to improve self-supervised depth and ego-motion estimation:

1. a novel per-pixel neural intensity transformation network to compensate for lighting changes;
2. a residual flow map for each pixel to correct the reprojection correspondences;
3. Using a Neighbour2Neighbour algorithm to denoise input images

**Issues:**

1. Generalisation: The method is trained and tested in the Oxford RobotCar dataset, with a good separation of training and testing routes. However, I am still a bit curious about whether the system trained in the Oxford Robotcar dataset can work well in other environments, day and night.
2. The assumption of constant illumination is an issue for photometric loss. Some existing works address this issue as well, but instead of using a motion network directly to regress the relative transformation, they formulate it as an optimisation problem. I believe they are also worth being mentioned in the related work section:
    1. Clement, Lee, and Jonathan Kelly. "How to train a cat: Learning canonical appearance transformations for direct visual localization under illumination change." IEEE Robotics and Automation Letters 3.3 (2018): 2447-2454.
    2. Bloesch, Michael, et al. "Learning meshes for dense visual slam." Proceedings of the IEEE/CVF International Conference on Computer Vision. 2019.
    3. Xu, Binbin, Andrew J. Davison, and Stefan Leutenegger. "Deep probabilistic feature-metric tracking." IEEE Robotics and Automation Letters 6.1 (2020): 223-230.
    4. Gridseth, Mona, and Timothy D. Barfoot. "Keeping an Eye on Things: Deep Learned Features for Long-Term Visual Localization." IEEE Robotics and Automation Letters 7.2 (2021): 1016-1023.
3. The idea of capturing the per-pixel changes in contrast (scale) and brightness (shift) looks interesting. I appreciate the theortical analysis in the supporting material to prove this assumption. However, it looks to me that there are too many simplifications involved to derive from the Phong illumination model to the proposed linear combination of scale and shift lighting change equation. In particular, the simplification of removing the reflection term looks a bit suspicious. I think a big portion of the lighting changes that occur at night is the reflection on roads and surrounding vehicles.
4. I am wondering why the proposed method does not utilise geometric consistency. The depth image $D_{t+1}$ from $I_{t+1}$ is not used in the pipeline.
4. The limitation section is a bit too general.

**Quality Of The Limitations Section:**

Additional details required

**Reviewer Expertise:**

4: The reviewer is confident but not absolutely certain that the evaluation is correct

**Robotics Focus:**

Highly relevant to robotics but no hardware experiments

**Strengths And Weaknesses:**

Strengths:
* The paper is well written and easy to follow.
* The lighting change decoder idea seems interesting and useful.
* The experiments are conducted on the public dataset and tested extensively at different combinations of day and night, with a good separation of training and testing routes.
* The result shows a consistent improvement compared to baseline methods.


Weaknesses:
* In general, the novelty in this paper is somewhat limited. It was not the first time that motion compensation and image denoising on the inputs are proposed for depth estimation and ego-motion learning.
* There are a few issues that could be treated as weaknesses. However, they may also be able to be addressed in the revision by the authors. Please take a look at the issues I listed below.

**Summary Of Recommendation:**

This paper may have a few limited theoretic novelties, but it shows well-conducted experiments and improvements compared to baseline methods. The paper can be improved if the mentioned issues are addressed.

---

> ### Author Response · Authors · 2022-08-26
> **Response to Reviewer BWdU**
>
> Q1: However, I am still a bit curious about whether the system trained in the Oxford Robotcar dataset can work well in other environments, day and night
>
> A1: We evaluated our model on the Once dataset (Mao J et al, Arxiv 21) to obtain the attached qualitative results. The images are taken from sequence number 340 of the medium unlabeled data split. It is important to note here that the lighting conditions of the Once dataset are very different from that of the RobotCar Dataset. Despite the differences, our model is still able to generate visually plausible depth maps with sharp structural details.
>
>
> Q2: Photometric vs Featuremetric loss literature:
>
> A2: Thanks for pointing out these existing works. We will update our related work section with these suggested papers as well as more recent work (Lindenberger et al. 2021, Sarlin et al. 2021, Chen et al. 2022, Stumberg et al. 2020a, Stumberg et al. 2020b) done in the direction of ‘featuremetric’ losses as opposed to photometric losses.
> Also, the reviewer rightly said that the photometric losses are known to have issues and features-based optimization is one of the alternatives. This has also been explored in the past for depth estimation, as in DeFeatNet, which our method outperforms. Our proposed approach uniquely highlights that it is possible to use photometric loss 24/7 as we are able to deal with the breaking points of photometric loss arising from temporal illumination consistency, static scene, and noise-free assumptions.
>
>
> Q3: The simplification of removing the reflection term looks a bit suspicious
>
> A3: We agree with the reviewer that many of the lighting changes that occur at night are based on reflected rather than directly emitted light (exceptions include e.g. flashing emergency lights etc.). It is also true that ignoring the Phong term in our derivation equates to ignoring specular reflections there, for reasons of simplicity. However, materials such as the tarmac on roads in any case mostly reflect light via diffuse rather than specular reflection, and diffuse reflection is captured by the Lambertian term of the Phong model, which we retain. The metallic surfaces of automobiles tend to reflect light via a mix of diffuse and specular reflection (please see e.g. Robert M Boynton, Encyclopedia of Physical Science and Technology, Third Edition); whilst our derivation does not treat the specular reflections involved, it does treat the diffuse ones. Moreover, it is worth bearing in mind that even though we ignore specular reflections in our derivation, they can nevertheless be captured by the lighting change images we predict.
>
> Q4: I am wondering why the proposed method does not utilize geometric consistency.
>
> A4: Geometric consistency, as per the literature (Bian et al., NeurlPS, 2019), is typically used to introduce scale and temporal consistency in the estimated depth maps as an auxiliary loss function. In this work, we focused on the core issues of the photometric loss assumptions. It should be possible to include geometric loss with our method to further improve results.
>
> Q5: The limitation section is a bit too general.
>
> A5: Thanks for pointing this out. We have now included the following two points in the limitations section which are more specific to our approach.
> 1. Our method assumes a linear relationship between pixel intensities and light intensity. This can be addressed by using the inverse camera-response function but will require a detailed analysis.
> 2. Even though our method handles the majority of the motion patterns in the image, it still suffers from estimating the motion of the objects moving with the same speed as the ego-vehicle in the same direction. Handling such objects remains future work.
>
> References:
>
> 1. Mao, Jiageng, et al. "One million scenes for autonomous driving: Once dataset." arXiv preprint arXiv:2106.11037 (2021).
> 2. Lindenberger, Philipp, et al. "Pixel-Perfect Structure-from-Motion with Featuremetric Refinement." Proceedings of the IEEE/CVF International Conference on Computer Vision. 2021.
> 3. Sarlin, Paul-Edouard, et al. "Back to the feature: Learning robust camera localization from pixels to pose." Proceedings of the IEEE/CVF conference on computer vision and pattern recognition. 2021.
> 4. Chen, Shuai, et al. "DFNet: Enhance Absolute Pose Regression with Direct Feature Matching." arXiv preprint arXiv:2204.00559 (2022).
> Von Stumberg, Lukas, et al. "Gn-net: The gauss-newton loss for multi-weather relocalization." IEEE Robotics and Automation Letters 5.2 (2020): 890-897.
> 5. Von Stumberg, Lukas, et al. "LM-Reloc: Levenberg-Marquardt based direct visual relocalization." 2020 International Conference on 3D Vision (3DV). IEEE, 2020.
> 6. Bian, Jiawang, et al. "Unsupervised scale-consistent depth and ego-motion learning from monocular video." Advances in neural information processing systems 32 (2019).

---

### Author Response · Authors · 2022-08-26
**Revised Paper and Supplementary Material**

**Comment:**

We have addressed all the comments from the reviewers in the revised versions of the main paper and supplementary material.

**Zip File:**

/attachment/db8b6089c7d3fa5c37bcc3cc4288e174f972c511.zip

---

### Meta-Review · Area_Chair_D8dU · 2022-08-02

**Recommendation:** Accept (Poster)
**Confidence:** 5

**Metareview:**

The paper proposes a new network for depth and ego-motion estimation. The main contribution is introducing three components in dealing with the photometric losses used in depth estimation methods. Majority of the reviewers agree on the contribution of the paper and acknowledge the novelty of the introduced three components in depth estimation. There are concerns on the generalization of the proposal method and several unclear parts in the paper.

The concerns from the reviewers have been successfully addressed during the rebuttal. The authors are encouraged to revise the final paper accordingly.

**Best Paper Nomination:**

No

---

> ### Author Response · Authors · 2022-08-26
> **Summary of changes**
>
> Thanks for your comments. Here, we summarize our response to the reviewers, including a brief description of new experiments conducted:
>
> 1. Generalization: We have now presented qualitative results on a new dataset under nighttime conditions, where depth maps visualization demonstrates the ability of our method to generalize to other cities.
>
> 2. Computational Analysis:  We have now included a computation time analysis to show the potential of our method in real-time applications on both GPU and CPU.
>
> 3. Text Clarifications: Our inline responses address a range of specific concerns raised by the reviewers for which we have provided clarifications, some of which will be included in the revised and final version of the paper (if accepted). These clarifications address concerns such as novelty, the relevance of the denoising module, photometric vs feature metric losses, modeling reflections, expanding limitations section, need for geometric consistency loss, egomotion estimation, hyperparameter selection, and general improvements of text, equations, and tables for improved readability and understanding.